# LMFusion: Adapting Pretrained Language Models for Multimodal Generation

**Weijia Shi**[*,w]   **Xiaochuang Han**[*,w]   **Chunting Zhou**[f]   **Weixin Liang**[s]   **Xi Victoria Lin**[f]

**Luke Zettlemoyer**[w,f]   **Lili Yu**[f]

[w]University of Washington   [f]FAIR at Meta   [s]Stanford University

swj0419@uw.edu   xhan77@uw.edu   liliyu@meta.com

## Abstract

We present LMFusion, a framework for empowering pretrained text-only large language models (LLMs) with multimodal generative capabilities, enabling them to understand and generate both text and images in arbitrary sequences. LMFusion leverages existing Llama-3's weights for processing texts autoregressively while introducing additional and parallel transformer modules for processing images with diffusion. During training, the data from each modality is routed to its dedicated modules: modality-specific feedforward layers, query-key-value projections, and normalization layers process each modality independently, while the shared self-attention layers allow interactions across text and image features. By freezing the text-specific modules and only training the image-specific modules, LMFusion preserves the language capabilities of text-only LLMs while developing strong visual understanding and generation abilities. Compared to methods that pretrain multimodal generative models from scratch, our experiments demonstrate that LMFusion improves image understanding by 20% and image generation by 3.6% while maintaining Llama-3's language capabilities. We also show that this framework can adapt existing vision-language models with multimodal generation ability.

## 1 Introduction

Over the past few years, we have seen significant progress in multimodal generative models capable of understanding and generating interleaved text and images in arbitrary sequences [1, 2, 3]. Models like Transfusion [4], Chameleon [5], and Unified-IO [6, 7] demonstrate the potential of unified architectures that seamlessly handle both image and text modalities. However, these models typically train from scratch, demanding significant computational resources to achieve proficiency across all modalities. The computational cost of mastering even a single modality is substantial—training a state-of-the-art text-only large language models (LLMs) like Llama-3 [8] requires training over 15 trillion tokens.

Given these computational demands, we investigate an alternative paradigm that reuses and adapts existing pretrained LLMs [9, 10, 11]. We address a fundamental research question: *How to preserve the text-only performance of pretrained LLMs while equipping them with visual understanding and generation abilities?* Our experiments show that naive finetuning of text-only LLMs on multimodal data leads to significant degradation of their language processing capabilities.

To address this challenge, we introduce **LMFusion**, a framework that enhances a pretrained text-only LLM, Llama-3 [8] with multimodal capabilities by building upon the recipe of Transfusion [4]. Drawing from recent and parallel work on modality separation [12, 13, 14, 15], LMFusion integrates the original Llama modules pretrained for language processing while introducing additional dedicated transformer modules for visual understanding and generation tasks. As shown in Figure 1, we

39th Conference on Neural Information Processing Systems (NeurIPS 2025).

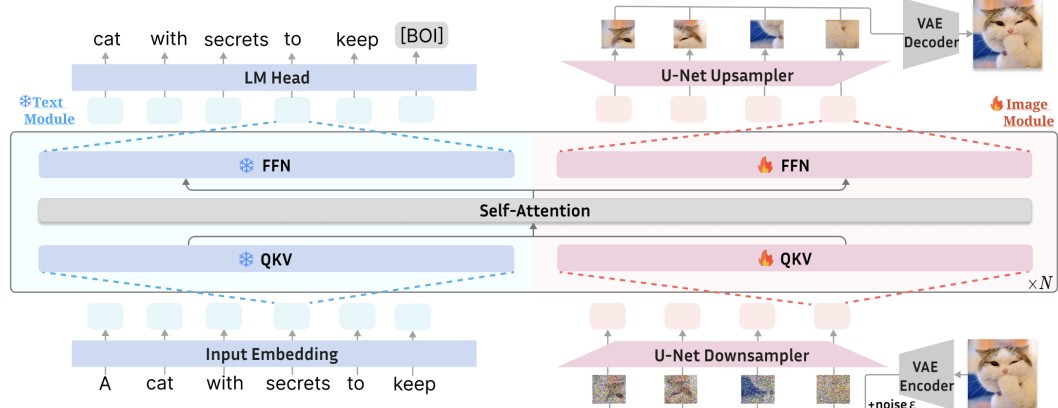

Figure 1: **Overview of LMFusion**. It uses modality-specific FFNs and QKV projections to process text and image data separately: the text "A cat with secrets to keep" goes to the text module , while the image patches of the cat goes to the image module . In the self-attention layer, text and image representations can attend to all previous contexts across the modality boundaries. Both modules are initialized from Llama-3, with the text module frozen to preserve language capabilities while the image module trained on image data. Layer normalization and residual connections are folded into the QKV and FFN modules. A special BOI token separates different modalities.

employ modality-specific query-key-value (QKV) projections and feed-forward networks (FFNs) to process text and image data separately while still allowing for cross-modal interactions in the joint self-attention layer. By freezing the text modules while finetuning the image modules, we preserve the language-only capabilities of pretrained LLMs while giving a head start to the learning of visual understanding and generation. Compared to pretraining multimodal generative models from scratch, this approach avoids the need to include text-only data in the training process, significantly reducing the computational demands.

To evaluate the effectiveness of our approach, we conduct comprehensive experiments comparing LMFusion with Transfusion in controlled settings. Specifically, we initialize our LMFusion architecture with a pretrained Llama-3 8B model [8] and continue training on the same image data as in Transfusion [4]. Compared to Transfusion, LMFusion achieves a 20% improvement in image understanding and 3.6% improvement in image generation. It also preserves Llama-3's text-only performance that outperforms Transfusion by 11.6%. Figure 2 presents images generated by LMFusion. Additionally, we further demonstrate that this framework can adapt existing vision-language models (e.g., LLaVA) with multimodal generation ability.

Through ablation studies, we analyze the key architectural decision for LMFusion: separating both self-attention and FFNs for different modality data while freezing weights for the pretrained language modality. We show that naive finetuning of the dense pretrained LLMs on multimodal data (*no separation*) leads to a catastrophic forgetting of their original language capabilities. Furthermore, deep separation proves to be more effective than shallow separation (*using modality-specific FFNs only*), with both approaches outperforming models with no separation.

## 2 Background: Transfusion

Transfusion [4] is a single unified multimodal model that is capable of text generation, image understanding, and image generation tasks, by jointly predicting next tokens in language and diffusing image representations. Given a multimodal input $(x^{txt}, x^{img})$, Transfusion jointly learns to do *language modeling* (§2.1) on $x^{txt}$ and *image diffusion* (§2.2) on $x^{img}$. Its architecture is same as a standard Transformer [16] with an additional U-Net [17] that projects image representations down and up before and after diffusion.

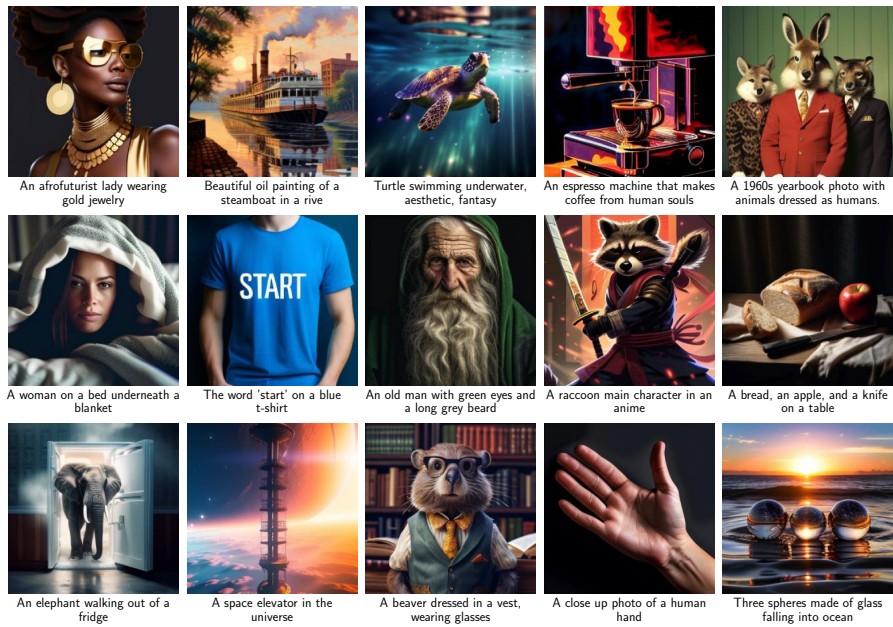

Figure 2: Generated images from LMFusion fine-tuned on aesthetically appealing images for improved quality.

## 2.1 Language Modeling

Given a sequence of discrete language tokens $\boldsymbol{x}^{txt} = x_1^{txt}, \ldots, x_N^{txt}$, a language model $\theta$ represents its joint probability by $P(\boldsymbol{x}^{txt}) = \prod_{i=1}^{N} P_\theta(x_i^{txt} \mid \boldsymbol{x}_{<i}^{txt})$. This formulation sets up an autoregressive task, where each token $x_i^{txt}$ is predicted based on its preceding tokens $\boldsymbol{x}_{<i}^{txt}$. The language model is learned by minimizing the cross-entropy between $P_\theta$ and the observed data distribution, which is commonly referred to as the LM loss:

$$\mathcal{L}_{\text{LM}} = \mathbb{E}_{x_i^{txt}}[-\log P_\theta(x_i^{txt} \mid \boldsymbol{x}_{<i}^{txt}, \boldsymbol{x}^{img})] \tag{1}$$

Optionally, if there exists image data preceding the language tokens (e.g., image-caption data), Transfusion adds the representation of $\boldsymbol{x}^{img}$ as additional condition to the objective. More details of representing $\boldsymbol{x}^{img}$ are presented below.

## 2.2 Image Diffusion

Given a raw image, Transfusion first encodes the image into a sequence of continuous latent representation $\boldsymbol{x}^{img}$ with a pretrained and frozen VAE tokenizer [18]. It then employs Denoising Diffusion Probabilistic Models (i.e., DDPM) to learn to reverse a gradual noise-addition process added in the forward process [19]. In the forward diffusion process, a Gaussian noise $\boldsymbol{\epsilon} \sim \mathcal{N}(\mathbf{0}, \mathbf{I})$ is added to the image representation $\boldsymbol{x}^{img}$ over $T$ steps, creating a sequence of noisy image representations $\boldsymbol{x}_0, \boldsymbol{x}_1, ..., \boldsymbol{x}_T$. Specifically, at each step $t$, the noisy image representation is given by:

$$\boldsymbol{x}_t^{img} = \sqrt{\bar{\alpha}_t} \boldsymbol{x}^{img} + \sqrt{1 - \bar{\alpha}_t} \boldsymbol{\epsilon} \tag{2}$$

Here $\bar{\alpha}_t$ follows a common cosine schedule [20]. In the reverse process, the diffusion model $\boldsymbol{\epsilon}_\theta(\cdot)$ with parameters $\theta$ learns to predict the added noise $\boldsymbol{\epsilon}$ given the noisy data $\boldsymbol{x}_t^{img}$ at timestep $t$ and a context $\boldsymbol{x}^{txt}$ that can include text prompts such as captions to the image diffusion: [1]

$$\mathcal{L}_{\text{DDPM}} = \mathbb{E}_{\boldsymbol{x}^{img}, t, \boldsymbol{\epsilon}}[\|\boldsymbol{\epsilon} - \boldsymbol{\epsilon}_\theta(\boldsymbol{x}_t^{img}, t, \boldsymbol{x}^{txt})\|_2^2] \tag{3}$$

The Transfusion architecture contains U-Net downsampler and upsampler to reduce the dimension of $\boldsymbol{x}^{img}$. The U-Net downsampler transforms the image into fewer patches before the main Transformer modules while the upsampler projects them back to the original dimension of $\boldsymbol{x}^{img}$ after the Transformer.

---

[1]Similar to $\boldsymbol{x}^{txt}$, this context can also include image representations $\boldsymbol{x}^{img}$ under an image editing setup. We omit it in the notation for simplicity.

## 2.3 Training Objective

During training, Transfusion is optimized to predict both the LM loss on the text input $x^{txt}$ and the diffusion loss on the image input $x^{img}$. These two losses are combined using a hyperparameter $\lambda$:

$$\mathcal{L}_{\text{Transfusion}} = \mathcal{L}_{\text{LM}} + \lambda \cdot \mathcal{L}_{\text{DDPM}} \tag{4}$$

# 3 LMFusion

One notable feature of Transfusion is that it has the same architecture as mainstream LLMs (e.g., Llama [21]) while being capable of text generation, image understanding, and image generation together, through an end-to-end training (Equation 4). [4] trains Transfusion from scratch using language-only and image-caption data. However, such training from scratch requires substantial computational resources, and its performance on language-only tasks still lags behind the pretrained, text-only LLMs. In this work, we aim to effectively adapt pretrained, text-only LLMs to handle image understanding and generation tasks. Specifically, we build on an open-weight LLM, Llama-3 [8], and continue training it with the Transfusion objectives to handle both modalities. Since Transfusion uses shared parameters for its language modeling and image diffusion objectives, the key challenge is to prevent Llama-3's strong text-only performance from dropping while optimizing for its new image capabilities.

## 3.1 Model Architecture

In response to the challenge above, we propose LMFusion, a framework that combines a pretrained, text-only Llama model with a dedicated image transformer for visual generation and understanding, enabling each modality to be processed through independent weights. By freezing the text modules while finetuning the visual modules, we preserve its language-only capabilities while giving the learning of visual understanding and generation a boost start.

LMFusion is a decoder-only model consisting of $N$ transformer layers. As shown in Figure 1, central to the design are the modality-specific attention layer and Feed-Forward Network (FFN), each handling only data from its corresponding modality. Without loss of generality, we describe LMFusion below in a configuration with a single transformer layer, folding residual connections and layer normalization directly into the self-attention and FFN. The inputs to the model are text tokens $x^{txt}$ and noisy image representations $x_t^{img} = \sqrt{\bar{\alpha}_t} x^{img} + \sqrt{1 - \bar{\alpha}_t}\epsilon$. We use blue for text-specific modules and red for image-specific modules.

**Input projection** The input text tokens $x^{txt}$ are projected by a linear embedding layer to a sequence of text hidden states $h_{\text{in}}^{txt}$. The noisy image $x_t^{img}$ are projected to a sequence of image representations $h_{\text{in}}^{img}$ via a U-Net downsampler.

$$h_{\text{in}}^{txt} = \text{Proj}_{\text{text}}(x^{txt}) \tag{5}$$

$$h_{\text{in}}^{img} = \text{UNet-Down}_{\text{img}}(x_t^{img}, t) \tag{6}$$

Then the text hidden states $h_{\text{in}}^{txt}$ or image hidden states $h_{\text{in}}^{img}$ are fed into the following attention layer.

**Modality-specific self-attention** We create separate attention matrices for each modality. Specifically, the text hidden states $h_{\text{in}}^{txt}$ and image hidden states $h_{\text{in}}^{img}$ are converted into their respective queries, keys, and values via separate $Q, K, V$ matrices. The pre-attention layer normalization is also modality-specific and is folded into the QKV functions.

$$h_{\text{Q}}^{txt}, h_{\text{K}}^{txt}, h_{\text{V}}^{txt} = \text{QKV}_{\text{text}}(h_{\text{in}}^{txt}) \tag{7}$$

$$h_{\text{Q}}^{img}, h_{\text{K}}^{img}, h_{\text{V}}^{img} = \text{QKV}_{\text{img}}(h_{\text{in}}^{img}) \tag{8}$$

We enable cross-modal attention by concatenating the queries, keys, and values from both image and text modalities into unified sequences. The attention-weighted values at text and image tokens are then projected back into the hidden state dimension using separate O weights for each modality.

$$h_{\text{O}}^{txt} = \text{O}_{\text{text}}(\text{softmax}(\frac{h_{\text{Q}}^{txt}[h_{\text{K}}^{img} \circ h_{\text{K}}^{txt}]^T + M}{\sqrt{d}})[h_{\text{V}}^{img} \circ h_{\text{V}}^{txt}]) \tag{9}$$

$$h_{\text{O}}^{img} = \text{O}_{\text{img}}(\text{softmax}(\frac{h_{\text{Q}}^{img}[h_{\text{K}}^{txt} \circ h_{\text{K}}^{img}]^T + M}{\sqrt{d}})[h_{\text{V}}^{txt} \circ h_{\text{V}}^{img}]) \tag{10}$$

where $\circ$ denotes concatenation. $M$ represents a hybrid attention mask same as in Transfusion [4] with a causal mask applied to text tokens and a bi-directional mask applied to image tokens. This design allows for self-attention within and across modalities, encouraging cross-modality integrations.

**Modality-specific feed-forward network**   After the attention layer, we employ modality-specific FFNs to process text and image data separately. The pre-FFN layer normalization is also modality-specific and is folded in the FFN functions.

$$\boldsymbol{h}_{\text{FFN}}^{txt} = \boxed{\text{FFN}_{\text{text}}}(\boldsymbol{h}_{\text{O}}^{txt}) \tag{11}$$

$$\boldsymbol{h}_{\text{FFN}}^{img} = \boxed{\text{FFN}_{\text{img}}}(\boldsymbol{h}_{\text{O}}^{img}) \tag{12}$$

**Output projection**   Finally, after $N$ layers of self-attention and FFNs, the resulting hidden states are projected either to logits in text via language model's output layer, or to predicted noise in image via a U-Net upsampler.

$$\boldsymbol{p}_{\text{logits}} = \boxed{\text{LM-Head}_{\text{text}}}(\boldsymbol{h}_{\text{FFN}}^{txt}) \tag{13}$$

$$\boldsymbol{\epsilon}_{\text{pred}} = \boxed{\text{UNet-Up}_{\text{img}}}(\boldsymbol{h}_{\text{FFN}}^{img}, t, \boldsymbol{h}_{\text{in}}^{img}) \tag{14}$$

Same as Transfusion, the output $\boldsymbol{p}_{\text{logits}}$ and $\boldsymbol{\epsilon}_{\text{pred}}$ are passed through the language modeling loss (Equation 1) and DDPM loss (Equation 3) respectively. All parameters in the text modules along with self-attention and FFN parameters in the image modules are initialized from the pretrained Llama model. During optimization, we ***decouple the learning rates*** for the text and image parameter groups: a text learning rate, $\eta_{\text{text}}$, is used for $\{\boxed{\text{Proj}_{\text{text}}, \text{QKV}_{\text{text}}, \text{O}_{\text{text}}, \text{FFN}_{\text{text}}, \text{LM-Head}_{\text{text}}}\}$, and an image learning rate, $\eta_{\text{img}}$, for $\{\boxed{\text{UNet-Down}_{\text{img}}, \text{QKV}_{\text{img}}, \text{O}_{\text{img}}, \text{FFN}_{\text{img}}, \text{UNet-Up}_{\text{img}}}\}$. To preserve the model's performance on text-only benchmarks, we use $\eta_{\text{text}} = 0$ (freezing text modules) for our main experiments and explore different configurations in §5.

## 4   Experiments

### 4.1   Training Setup

**Data**   Following Transfusion [4], we use the same collection of 380M Shutterstock image-caption data, where each image is center-cropped and resized to $256 \times 256$ pixels. We order the captions before images (i.e., emphasizing image generation conditioned on texts) 80% of the time, and order the images before captions for the rest.

**Model Details**   For image tokenization, we use the same VAE encoder[2] as Transfusion to compress an image of $256 \times 256$ pixels into a $32 \times 32 \times 8$ tensor. These tensors are then passed into a 2-block U-Net downsampler [17] to further reduce dimensions, resulting in a sequence of 256 patches (tokens). Both text-specific and image-specific Transformer modules are initialized from the pretrained Llama-3 8B model [8]. The U-Net downsampler and a corresponding U-Net upsampler, totaling 0.27 billion parameters [4], are trained from scratch. Like Transfusion, LMFusion uses a maximum context length of 4096 tokens.

**Optimization**   In our main experiments, to preserve the language-only performance, we freeze the text modules ($\eta_{\text{text}} = 0$) while training only the image modules using an AdamW optimizer ($\beta_1 = 0.9$, $\beta_2 = 0.95$, $\epsilon = 1 \times 10^{-8}$) with a learning rate $\eta_{\text{image}} = 1 \times 10^{-4}$. The learning rate follows a cosine decay schedule with a 4000-step warmup period before gradually decreasing to $1.5 \times 10^{-5}$. The model is trained using 128 H100 GPUs over 4 days.

### 4.2   Evaluation Setup

We compare our model with both the original Transfusion 7B model trained from scratch [4] and the Transfusion model initialized from the same LLaMA model. [3] The original Transfusion was trained for 250K steps on 0.25T language-only tokens (text data) and 0.25T image-captions tokens (image data). Since we freeze and reuse the text module from existing text-only models during training, we can exclude text data from our training process while maintaining language capabilities. In the first

---

[2]`https://huggingface.co/stabilityai/sd-vae-ft-mse`

[3]Transfusion 7B and Llama-3 8B have the same Transformer sizes. The size difference is due to the different vocabularies, which affects input and output embedding layers.

| Model | Language-only Evaluation | | | Image Understanding | Image Generation (without \| with CFG) | |
|---|---|---|---|---|---|---|
| | HellaSwag↑ | SIQA↑ | WinoGrande↑ | CIDEr ↑ | FID ↓ | CLIP ↑ |
| Llama 3 | 60.0 | 48.1 | 72.8 | – | – | – |
| Transfusion | 51.0 | 42.3 | 64.3 | 32.0 | 14.4 \| 8.70 | 22.1 \| 24.4 |
| LMFusion (0.5x FLOPs) | 60.0 | 48.1 | 72.8 | 38.3 | 13.9 \| 8.75 | 22.0 \| 24.4 |
| LMFusion (1.0x FLOPs) | 60.0 | 48.1 | 72.8 | 38.4 | 14.0 \| 8.61 | 22.1 \| 24.4 |

Table 1: **Results across text-only benchmarks, image understanding and image generation.** LMFusion preserves Llama-3's text performance while adding strong image understanding and generation capabilities. Image generation results are obtained without classifier-free guidance or with a CFG factor of 1.55. **Additionally, we show detailed analyses of our full LMFusion recipe vs. the vanilla Transfusion recipe initialized from Llama-3 in Figure 4 and Figure 5.**

configuration, we use the amount of 0.25T image data in Transfusion while leaving out the text data (approximately half the total FLOPs of Transfusion), while in the second configuration, we match Transfusion's total FLOPs. Following Transfusion, we evaluate LMFusion on language-only, image understanding, and image generation tasks.

**Language-only**: We evaluate the model's language abilities using four tasks from the standard Llama evaluation suite [8], including Hellaswag [22], PIQA [23], SIQA [24], and WinoGrande [25]. We report accuracy on these benchmarks. **Image Generation**: For evaluating image generation, we use the MS-COCO benchmark [26]. We generate images for 30K randomly selected prompts from the validation set and measure the Frechet Inception Distance (FID) [27] and CLIP scores [28]. Our image generation results include versions obtained without classifier-free guidance and with a CFG coefficient of 1.55 or 1.6. **Image Understanding**: We evaluate the models' ability to generate image descriptions using the test split of MS-COCO [26], reporting CIDEr scores [29].

## 4.3 Results

Table 1 compares two variants of LMFusion against Transfusion. On language-only benchmarks, LMFusion keeps the strong performance of Llama-3 since we freeze all text modules. For image understanding, LMFusion substantially surpasses Transfusion, with a 20% improvement. In image generation tasks, LMFusion also shows superior results in both FID and CLIP scores. Furthermore, in §5, we show from Figure 4 that LMFusion outperforms Transfusion initialized from Llama-3 (i.e., dense model with no separation) during the training process. In Figure 3, we benchmark the performance of LMFusion and Transfusion throughout the training.[4] We observe a consistent advantage of LMFusion over Transfusion during the entire training, for image captioning and generation. These results suggest that LMFusion effectively leverages the pretrained language modules from Llama while developing strong image abilities. Although LMFusion has twice as many parameters as Transfusion, it uses same training FLOPs since only half of the parameters are activated for each input token from an arbitrary modality.

## 5 Analysis

Central to LMFusion is our modality separation techniques, which employs the design of modality-specific modules and decoupled learning rates for language and image modules. Our architectural ablation (§5.1) demonstrates the importance of the design for maintaining model performance across both modalities. We further showcase that this recipe could be used for adapting pretrained vision-language models.

### 5.1 Architecture Ablations

#### 5.1.1 Experimental Design

To evaluate different design choices, we conduct ablation studies using small-scale variants of LMFusion. Our analysis focuses on the impact of modality separation by comparing three designs:

---

[4]For the image generation results plotted throughout the training, we use a smaller subset of 5K prompts and without classifier-free guidance.

|  |  | Image Und. | | | | Image Gen. |
| Model | Base LLM | MMMU ↑ | ChartQA ↑ | RealWorldQA ↑ | MME-P ↑ | FID ↓ |
|---|---|---|---|---|---|---|
| EMU-3 | – | 31.6 | 51.8 | 57.4 | – | 12.8 |
| Show-O | Phil-1.5 1.3B | 27.4 | – | – | 1435.7 | 9.2 |
| Janus | DeepSeek 1.3B | 30.5 | – | – | 1338.0 | 8.5 |
| Chameleon | – | 28.4 | 0.0 | 19.6 | – | 26.7 |
| MetaMorph | LLaMA-3.1 8B | **41.8** | 37.1 | 58.3 | – | 11.8 |
| Transfusion | – | – | – | – | – | 8.7 |
| LLaVAFusion | LLaVA-Next 8B | 41.7 | **69.5** | **60.0** | **1603.7** | **8.2** |

Table 2: **Comparison of multimodal models across image understanding and generation capabilities.** Models are evaluated on various image understanding benchmarks and image generation quality (FID). The models without base LLM are pretrained from scratch.

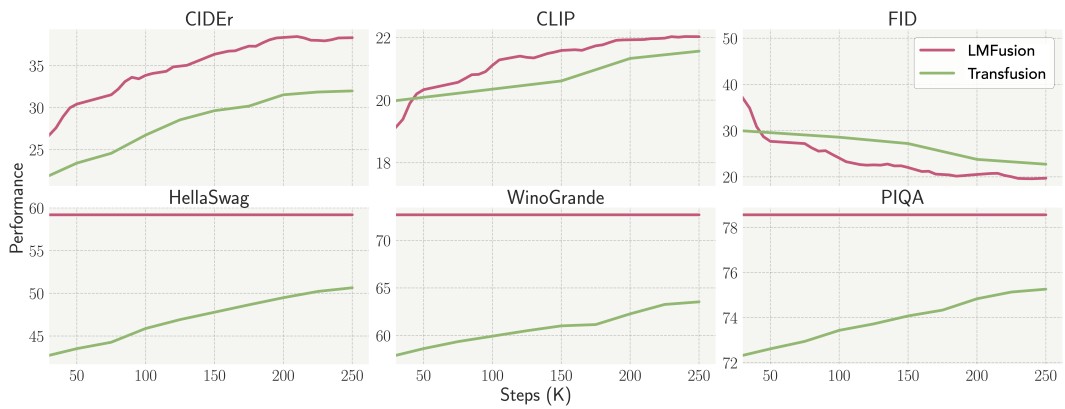

Figure 3: **Evaluation of LMFusion and Transfusion during training.** LMFusion keeps the text performance of Llama throughout training, while achieving better image understanding ability (CIDEr) and image generation quality (CLIP, FID).

(1) *no separation* (a single dense model), (2) *shallow separation* (using modality-specific FFNs only), and (3) *deep separation* (using both modality-specific FFNs and attention mechanisms, our final LMFusion).

**No separation (dense model)** We begin our experiments with the dense Llama-3 8B model trained using the Transfusion recipe. This dense model maintains a unified structure where most components are shared across modalities (a single set of QKV, O and FFN process both texts and images), with the exception of U-Net upsampler and downsampler. For training, we use a text learning rate ($\eta_{\text{text}}$) for the components initialized from the text-only LLM { Proj$_{\text{text}}$, QKV, O, FFN, LM-Head$_{\text{text}}$ }, and an image learning rate $\eta_{\text{img}}$ for { UNet-Down$_{\text{img}}$, UNet-Up$_{\text{img}}$ }. To investigate the impact of learning rate decoupling, we experiment with various learning rate ratios $\frac{\eta_{\text{text}}}{\eta_{\text{image}}} \in \{0, 0.1, 1\}$, with a constant image learning rate $\eta_{\text{image}} = 1 \times 10^{-4}$, the same as the main experiments. A ratio of 1 represents standard continual pretraining where all components share the same learning rate, while a ratio of 0 indicates a complete freezing of text-related components.

**Shallow separation (modality-specific FFNs only)** We explore a simplified variant of LMFusion that separates only FFNs into text-specific and image-specific modules—a common approach in mixture-of-experts architectures [3, 30]. In this setup, we use a single shared attention mechanism (QKV, O) for processing both image and text data. For training, we employ separate learning rates: $\eta_{\text{text}}$ for text-related components { Proj$_{\text{text}}$, QKV, O, FFN$_{\text{text}}$, LM-Head$_{\text{text}}$ } and $\eta_{\text{img}}$ for image-related components { Unet-Down$_{\text{img}}$, FFN$_{\text{img}}$, Unet-Up$_{\text{img}}$ }. We experiment with various learning rate ratios $\frac{\eta_{\text{text}}}{\eta_{\text{image}}} \in \{0, 0.1, 1\}$.

**Deep separation (modality-specific FFNs and attention)** Our LMFusion, as described in section 3, represents a deep separation design where both FFNs and attention mechanisms are modality-specific.

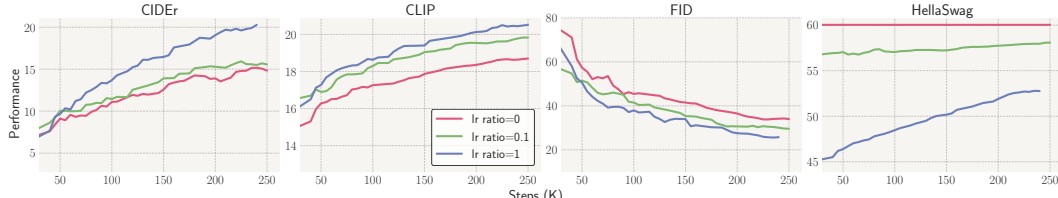

Figure 4: **Performance of naive Llama-3 finetuning (no separation) with varying lr ratio $\frac{\eta_{\text{text}}}{\eta_{\text{image}}}$.**
When directly finetuning the Llama-3 model for multimodal generation, using the same learning rate for both text and image components (lr ratio = 1) substantially reduces its text-only performance. Lowering the learning rate for the text component relative to the image component (lr ratio < 1) helps preserve language performance but slows down the acquisition of multimodal abilities.

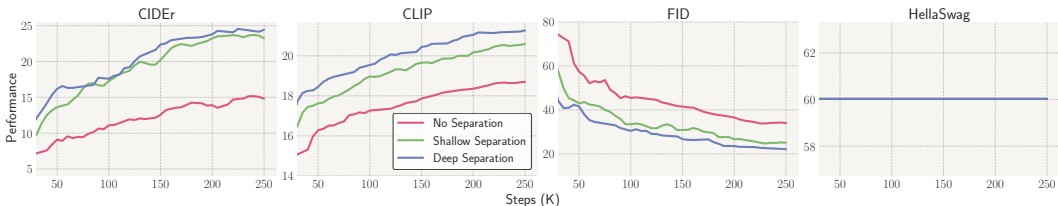

Figure 5: **Performance of *no separation* (dense model), *shallow separation* (modality-specific FFNs only), and *deep separation* (modality-specific FFNs and attention) when text modules are frozen.** Deep modality separation outperforms shallow separation and no separation.

While our primary configuration freezes text modules during training, we also analyze the impact of different learning dynamics by varying the learning rate ratio $\frac{\eta_{\text{text}}}{\eta_{\text{image}}}$ across $\{0, 0.1, 1\}$.

In the ablation study, all models are trained for 250K training steps with a sequence length of 4,096 tokens and a batch size of 250K tokens. The training data comprised 0.03T text-only tokens and 0.03T image-caption tokens. All other hyperparameters remained consistent with those employed in our main experiments.

### 5.1.2 Results

**Naive finetuning of dense pretrained LLMs for multimodal generation compromises their original language capabilities.** When directly finetuning Llama-8B (no separation) using the Transfusion recipe, we observe significant performance trade-offs between image and text capabilities (Figure 4). With equal learning rates for text and image components ($\frac{\eta_{\text{text}}}{\eta_{\text{image}}} = 1$), the model shows continuous improvement in image understanding and generation. However, this comes at a substantial cost to language capabilities, with performance on HellaSwag dropping by 15% initially. While language performance improves during training, it never recovers to the original Llama-3 model's level, maintaining a persistent 7% gap.

To mitigate this issue, we explore setting $\frac{\eta_{\text{text}}}{\eta_{\text{image}}} < 1$, which allows us to train image-specific modules (U-Nets) with a regular learning rate while preserving text abilities using a smaller learning rate for the general Transformer components. Figure 4 shows this improves language-only benchmark performance, reducing the gap from 7% to 2% when the ratio is 0.1. However, for dense models, this improvement comes at the cost of consistently reduced image capabilities. Overall, while learning rate decoupling offers some mitigation to the text performance drop, training dense pretrained LLMs without modality separation remains suboptimal.

**Deep Modality Separation Outperforms Shallow Separation.** In Figure 5, we compare three architectures: no separation (dense), shallow separation (modality-specific FFNs only), and deep separation (modality-specific FFNs and attention). We set $\frac{\eta_{\text{text}}}{\eta_{\text{image}}} = 0$ (freezing the text module) across all models to maintain Llama-3's text performance. Both separation approaches significantly outperform the dense model on all image benchmarks. While shallow separation performs slightly worse on image understanding, the performance gap widens notably in image generation tasks.

Additionally, deep separation with $\frac{\eta_{\text{text}}}{\eta_{\text{image}}} = 0$ has the same amount of *tunable* parameters as no separation with $\frac{\eta_{\text{text}}}{\eta_{\text{image}}} = 1$. Despite the intrinsic advantage of modality separation for text-only tasks,

for image understanding and generation, we still observe that deep separation (blue curve in Figure 5) are better than no separation (blue curve in Figure 4). These results show that modality separation is crucial for adapting pretrained language-only LLMs for multimodal generation.

## 5.2 LLaVAFusion: extending LMFusion to vision-language models

LMFusion continues training the language-only pretrained LLM Llama with the Transfusion recipe. Can this recipe be extended to on vision-language models (VLMs) such as LLaVA [31, 32] and Qwen-VL [33] as well? In this section, we extend the recipe of LMFusion to VLMs, preserving their multimodal understanding capabilities while introducing image generation abilities. Specifically, we build on LLaVA-NeXT [32], freezing its transformer parameters and integrating a dedicated, image-specific transformer module trained in parallel. We use the same data and model settings as LMFusion. We refer to this new model as LLaVAFusion and demonstrate its image understanding performance on MMMU [34], MME-Perception [35], ChartQA [36], and RealWorldQA [37], as well as its image generation results. For baselines, we compare LLaVAFusion against EMU-3 [38], Show-O [39], Janus [40], Chameleon [41], MetaMorph [42], and Transfusion [4]. As shown in Table 2, LLaVAFusion LLaVAFusion demonstrates strong performance in both image understanding and generation when compared to other unified multimodal LMs. This demonstrates that LMFusion is promising as an extension not only to language-only LLMs but also to VLMs, enhancing the multimodal generation capabilities in both cases.

## 6    Related Work

**Unified Models for Multimodal Generation**: Recent work has extensively explored unified frameworks for multimodal generation, including text generation, image understanding, and image generation. While texts are commonly represented as discrete tokens across models, approaches to representing images—especially for image generation—vary significantly. For instance, methods in [6, 43, 7, 5, 44, 11], represents images using vector-quantized discrete tokens [45, 46, 47]. An alternative method, adopted by [48, 49], employs continuous embeddings that require a separate diffusion model for decoding. In this work, we build upon Transfusion [4], which integrates autoregressive generation for texts with diffusion for images within a single model. **Model Sparsity**: Model sparsity through Mixture of Experts (MoE) [50, 30, 51, 52] has proven highly effective in improving LLM training efficiency. This approach has recently been extended to multimodal models [12, 53, 54, 55], particularly to address potential conflicts between different modalities. For example, recent efforts [13, 3, 56, 57] replace standard Transformer FFNs with modality-specific experts, enabling separate processing paths for different modalities. Our work takes this concept further by using modality-specific attention mechanisms. Concurrent work [15, 14] demonstrates the effectiveness of this deeper separation in multimodal pretraining and image generation. **Reuse of LLMs in Multimodal Training**: Based on the strong language capabilities of LLMs, some recent models on multimodal generation initializes their models from pretrained, language-only LLMs. For example, [9, 10, 1, 44, 11, 58] continued training upon the weights of language-only LLMs [21] or vision LLMs without visual generation capabilities [33]. The main focus of our work is to effectively reuse pretrained LLMs for multimodal generation, particularly with the Transfusion recipe, without any compromise on the LLMs' existing text-only capabilities.[5]

## 7    Conclusion

We present LMFusion, a framework designed to equip LLMs with multimodal generative capabilities. By using Llama-3 for text generation and integrating parallel transformer modules for image diffusion, LMFusion efficiently reuses compute of pretrained LLMs. LMFusion's modular design enables independent developments of language and vision modules, de-risking the complexities associated with a large-scale, joint-modality pretraining. In this work, we reuse only the pretrained language components, which still requires substantial compute to train the image generation module from scratch. Future work could explore reusing pretrained image generation components as well.

---

[5]Concurrent to our work, [15] tackles multimodal generation via a joint attention mechanism between a DiT structure [59] for images and a frozen Llama-3 [8] for texts.

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
