# OpenReview forum: "LMFusion: Adapting Pretrained Language Models for Multimodal Generation"
_NeurIPS.cc/2025/Conference — NeurIPS 2025 poster_

### Official Review · Reviewer_LX3F · 2025-06-29

**Clarity:** 4
**Significance:** 3
**Originality:** 2
**Rating:** 5
**Confidence:** 4

**Summary:**

This work introduces LMFusion, a method to enable pre-trained language models to generate and understand images in addition to text. LMFusion achieves this by starting from a pre-trained language model (Llama-3) and extending it with a parallel transformer module that is also initialized from the same Llama-3 model, with each module possessing separate modalities-specific modules.

The final version of LMFusion:
* is more compute efficient than comparable methods pretraining multimodal generative models from scratch
* achieves strong performance in image generation and understanding
* retains the capabilities of the original LLM used to seed the model

Finally, the same methodology is validated on existing VLMs.

**Questions:**

1. In the Transfusion paper, it is explained that in the setting with the image coming before text, the noising process on the images is only limited to half of the noise schedule, as fully noising the images (which condition the subsequent text) has a clearly negative effect. Did you do the same operation here?
2. Is a full copy of Llama completely necessary as image module? Would it be possible to decouple the parameter count for the 2 modalities instead of having the same?
3. Mostly out of curiosity, I was wondering if you had experimented inference with different kind of multi-modal sequences (e.g. multiple instances of text-image-text-image)

**Ethical Concerns:**

["NO or VERY MINOR ethics concerns only"]

**Final Justification:**

The authors have responded to points and questions raised by myself and other reviewers. I believe my original assessment is appropriate for this work.

**Limitations:**

yes

**Quality:**

3

**Strengths And Weaknesses:**

Strengths:
- This work is clearly written and the presented methodology is both well motivated (re-using available pre-trained weights can severely cut the amount of FLOPs needed) and justified by prior literature in the field
- The findings are corroborated by a good amount of evaluations for both the text and image domains (including both image generation and understanding)
- The components of the final model (e.g. the relative learning rates, the type of separation) are also described clearly and their choice is also based on ablations over each individual component

Weaknesses:
- As a minor comment: while the work is well executed and of high significance, it does not seem particularly original

---

> ### Author Rebuttal · Authors · 2025-07-31
>
> We thank the reviewer for your time and very insightful feedback! We answer each of your questions below:
>
> > In the Transfusion paper, it is explained that in the setting with the image coming before text, the noising process on the images is only limited to half of the noise schedule, as fully noising the images (which condition the subsequent text) has a clearly negative effect. Did you do the same operation here?
>
> Yes, to ensure a fair and direct comparison, we closely followed the training protocol of Transfusion, which includes using the limited noise schedule when an image serves as a condition for subsequent text generation. Future work may further explore other upper bound values of the noise schedule for images as conditions.
>
> > Is a full copy of Llama completely necessary as an image module? Would it be possible to decouple the parameter count for the 2 modalities instead of having the same?
>
> This is an excellent point. A full copy is not strictly necessary. Our framework is flexible and could accommodate differently sized image modules. The cross-attention mechanism allows for effective interaction between modules, and a simple linear projection could be added to merge representations if the hidden dimensions were different. We believe this is a promising direction for future work to optimize the parameter-performance trade-off.
>
> > Mostly out of curiosity, I was wondering if you had experimented inference with different kind of multi-modal sequences (e.g. multiple instances of text-image-text-image)
>
> Yes, we experimented with multi-modal sequences for image editing tasks. Due to the page limit of the submission, we did not include these results but will incorporate the following paragraph and qualitative examples in the final version of the paper.
> “We fine-tuned LMFusion on a dataset of 8K image editing examples, each consisting of an original input image, a prompt detailing the desired edit, and a resulting image that reflects the specified changes. We then applied the fine-tuned LMFusion to input images and editing prompts from the MagicBrush [1] test set. Qualitative results show that LMFusion performs effectively in these image-editing scenarios, complementing its strong capabilities in text-only, image understanding, and image generation tasks."
>
> [1] MagicBrush: A Manually Annotated Dataset for Instruction-Guided Image Editing

---

> > ### Comment · Reviewer_LX3F · 2025-08-07
> >
> > Thank you for the response.
> > The authors have addressed all the points I raised. Given they were mostly about exploration of open questions and not limitations of the paper and I already gave a positive feedback on the work, I'm confirming the rating of 5.

---

> ### Author Response · Authors · 2025-08-04
>
> Thank you for the feedback and suggestion. Please let us know if you have any further questions. We would be happy to address them.

---

### Official Review · Reviewer_edtZ · 2025-07-02

**Clarity:** 3
**Significance:** 3
**Originality:** 3
**Rating:** 5
**Confidence:** 4

**Summary:**

This paper proposes LMFusion, a method that enables image generation capability for LLMs or VLMs. To address the forgetting problem caused by joint multimodal training, LMFusion designs independent parameters for the image and text modalities, completely decoupling the two modalities. Experimental results show that this approach can introduce image generation capability without compromising language ability.

**Questions:**

1. **LLaVAFusion**
In section 5.2, the author proposed LLaVAFusion, empowering LLaVA with image generation capability.
Based on my understanding, the authors feed the image tokens used for visual understanding in the instruction into the original model, while the image tokens used for image generation are sent to the newly decoupled attention and FFN layers.
Have the authors tried feeding the visual understanding tokens into the newly established decoupled attention and FFN layers as well? This would be more consistent with the modality-specific claim made by the authors.

**Ethical Concerns:**

["NO or VERY MINOR ethics concerns only"]

**Final Justification:**

The author's response has addressed most of my concerns. I believe LMFusion is a very forward-looking work and should be accepted.

**Limitations:**

Yes

**Quality:**

3

**Strengths And Weaknesses:**

### Strengths

1. **Paper writing**
The paper is well written, with a clear motivation and methodology. The figures and tables in the experimental section are also presented very clearly.

2. **Reasonable idea.**
Separating parameters between different modalities is a very reasonable idea, and it has indeed achieved good results.

### Weakness

1. **Parameter size.**
Introducing modality-specific layers significantly increases the model size. According to the paper, the attention and FFN layers are duplicated for each modality, effectively doubling the parameter count and turning an 8B model into a 16B model. This is my major concern in this paper. If adding more modality (like audio, 3D), the number of parameters will continue to grow accordingly.

2. **Unfair comparision**
As mentioned above, LMFusion is a 16B model. Therefore, the authors should make a comparison with models of similar size (around 16B).

3. **Speed and Memory**
It would be helpful if the authors could provide a detailed comparison of training and inference time, FLOPs, as well as GPU memory consumption, for models that share parameters and separate parameters for different modalities.

---

> ### Author Rebuttal · Authors · 2025-07-31
>
> We thank the reviewer for your time and constructive feedback! We address each of your comments below:
>
> > Parameter size. Introducing modality-specific layers significantly increases the model size. According to the paper, the attention and FFN layers are duplicated for each modality, effectively doubling the parameter count and turning an 8B model into a 16B model. This is my major concern in this paper. If adding more modality (like audio, 3D), the number of parameters will continue to grow accordingly.
>
> While LMFusion has twice the total parameters of Transfusion, the active parameters remain unchanged for any given input, **resulting in the same inference FLOPs as Transfusion.** Our modality-specific routing ensures that for any given input token, only the corresponding modules are activated: text tokens are processed by the frozen text modules, and image tokens are processed by the trainable image modules. We acknowledge that the total model size would grow with additional modalities, but the active model size remains constant.
>
>
>
>
> > Unfair comparision As mentioned above, LMFusion is a 16B model. Therefore, the authors should make a comparison with models of similar size (around 16B). Speed and Memory It would be helpful if the authors could provide a detailed comparison of training and inference time, FLOPs, as well as GPU memory consumption, for models that share parameters and separate parameters for different modalities.
>
> We respectfully disagree that this constitutes an unfair comparison, as the inference FLOPs of LMfusion are identical to Transfusion while training FLOPs are one-third less than Transfusion, as shown in Figure 3. **Despite having twice the total parameters, LMFusion is computationally equivalent to Transfusion during inference and significantly more efficient during training.**  We thank the reviewer for the suggestion and will include detailed comparisons in the final version of the paper.
>
>
> > LLaVAFusion In section 5.2, the author proposed LLaVAFusion, empowering LLaVA with image generation capability. Based on my understanding, the authors feed the image tokens used for visual understanding in the instruction into the original model, while the image tokens used for image generation are sent to the newly decoupled attention and FFN layers. Have the authors tried feeding the visual understanding tokens into the newly established decoupled attention and FFN layers as well? This would be more consistent with the modality-specific claim made by the authors.
>
> This is a great point. In the interleaved setup, visual tokens are routed based on their intended use: visual tokens for image generation are processed by the image generation tower, while tokens for visual understanding are processed by the VLM tower. We have not yet explored routing visual understanding tokens through the image generation tower, but this would be interesting future work. We will include a discussion of this approach in the future work section.
>
> We hope our responses have addressed the reviewer's concerns. Please let us know if you may have any questions.

---

> ### Author Response · Authors · 2025-08-04
>
> Thank you for the feedback and suggestion. Please let us know if you have any further questions. We would be happy to address them.

---

> > ### Comment · Reviewer_edtZ · 2025-08-06
> >
> > Thank you for the author's response. I think LMFusion is a great work, and there have been many excellent follow-up works (e.g., Bagel). However, I still have some concerns regarding this work.
> >
> > 1. I think that the structure of LMFusion is task-specific rather than modality-specific. I think the authors can clarify this point briefly in the paper.
> > 2. I also have some concerns regarding multi-turn dialogue scenarios, such as first generating an image and then performing VQA (understanding) or editing (generation) based on this generated image. In such cases, I think it is difficult for the structure of LMFusion, or more precisely LLaVAFusion, to handle. In different turns, the image may need to go through different experts, which would require additional computational resources.

---

> > > ### Author Response · Authors · 2025-08-07
> > >
> > > Thank you for your response and we highly appreciate both of your comments.
> > >
> > > > I think that the structure of LMFusion is task-specific rather than modality-specific. I think the authors can clarify this point briefly in the paper.
> > >
> > > We will clarify in the next version of our paper that our modality-specific terminology, especially in the LlavaFusion setup, means more towards "meta-task-type"-specific (language-only, image understanding, image generation tasks).
> > >
> > > > I also have some concerns regarding multi-turn dialogue scenarios, such as first generating an image and then performing VQA (understanding) or editing (generation) based on this generated image. In such cases, I think it is difficult for the structure of LMFusion, or more precisely LLaVAFusion, to handle. In different turns, the image may need to go through different experts, which would require additional computational resources.
> > >
> > > Indeed in a production scenario of multi-turn dialogues, mixing t2i, i2t, and image editing, the image may need to go through different experts. However, we don't think this is a fundamental flaw or drawback of our framework. Additionally, this could be relatively alleviated over time as more unified tokenizers unifying perception and reconstruction emerge. We appreciate the comment and the forward thinking.

---

> > > > ### Comment · Reviewer_edtZ · 2025-08-07
> > > >
> > > > Thank you for the author's response. Overall, I think LMFusion is a very good work, and I will raise my score to 5.

---

### Official Review · Reviewer_FnMX · 2025-07-02

**Clarity:** 3
**Significance:** 2
**Originality:** 2
**Rating:** 2
**Confidence:** 3

**Summary:**

This paper introduces LMFusion, a unified model that extends pretrained text-only large language models (LLMs), specifically Llama-3, with multimodal generative capabilities. LMFusion achieves this by integrating modality-specific modules for vision (e.g., separate FFNs and QKV projections for image inputs) while keeping the original language model frozen to preserve its textual capabilities. The architecture supports cross-modal interactions through shared self-attention layers and uses diffusion-based modules for image generation. Compared to the unified model Transfusion, the approach demonstrates strong gains in image understanding and generation, while maintaining superior language modeling performance. Ablation studies validate the importance of modality separation and freezing.

**Questions:**

1. Could you share how LLaVAfusion performs on standard image understanding benchmarks (e.g., VQAv2, MMMU) compared to the base LLaVA-Next-8B?
Additionally, while not directly claimed in the paper, I'm curious:
Have you observed any effects—positive or negative—of jointly training for both image generation and image understanding on the model's performance in understanding tasks?
Any insight or ablation in this direction would help clarify the broader implications of your approach.

2. The paper observes that naive finetuning of dense pretrained LLMs for multimodal generation compromises their language capabilities, which motivates the modality separation approach in LMFusion. However, this might also suggest that the proposed architecture—by strictly separating modalities and freezing the language model—does not fully align image and text representations in a shared semantic space.
How do the authors view this trade-off between preserving language ability and achieving deep cross-modal alignment?
Are there any signs that the current setup limits joint reasoning or compositional understanding across modalities?

**Ethical Concerns:**

["NO or VERY MINOR ethics concerns only"]

**Final Justification:**

Most of my concerns remain unaddressed after the rebuttal, and the authors’ response has raised additional issues:

1. *Implementation Discrepancy*: The rebuttal highlights a key discrepancy between the implementations of LMFusion and LLavaFusion. Specifically, LMFusion leverages a learned image generation encoder for image understanding, whereas LLavaFusion avoids this due to observed performance degradation. This inconsistency raises concerns about the general effectiveness of the proposed alignment strategy between image generation and pre-trained LLMs in the current framework.

2. *Framework Limitation and Imbalance in Focus*: The paper presents the finding that a deep modality separation is required to prevent image generation from degrading language and image understanding performance. While this is an interesting observation, it also reflects a fundamental limitation of the framework in handling diverse multimodal tasks. Furthermore, as a work on multimodal generation, the paper disproportionately focuses on language and image understanding where their performance is inherited from their base frozen LLM/MLLM. The paper pays relatively little attention and analysis devoted to image generation. From Table 1, image generation performance shows only marginal improvement under the same LLM backbone, and the paper fails to explore the reasons or implications behind this. In the rebuttal, the authors promise to provide more analysis in the revision, but they do not present any during the rebuttal phase, which leaves me unconvinced about this commitment.

3. *Lack of Broader Comparison*: The authors do not compare their method with proprietary unified models such as GPT-4o or Gemini, nor with specialist image generation models like Flux or Stable Diffusion 3. This omission limits the reader’s understanding of how the proposed method fits into the broader landscape of multimodal generation research.

Overall, I find this work to be a less successful attempt at integrating image generation capabilities into pre-trained LLMs. The paper does not offer sufficiently novel insights or contributions to the field, particularly in its current form. While I acknowledge the effort of this work, especially the analysis of modality separation in multimodal generation, I remain skeptical about whether it meets the threshold for a NeurIPS main conference paper. I believe this work would be more appropriately presented in a workshop setting.

**Limitations:**

The paper does not include an explicit discussion of its limitations. A limitations section would be valuable, particularly to clarify:

1. How the proposed approach compares conceptually and practically to specialist large multimodal models (e.g., Qwen-2.5-VL, GPT4o, Gemini) in terms of image-text understanding and joint reasoning.

2. How LMFusion's diffusion-based image generation compares to specialized diffusion models (e.g., Flux, Stable diffusion 3), both in terms of quality and flexibility.

A discussion on these points would help position LMFusion more clearly in the current landscape and provide insight into the trade-offs between generality, specialization, and efficiency.

**Paper Formatting Concerns:**

The paper format looks good to me.

**Quality:**

2

**Strengths And Weaknesses:**

Strengths:

1. Clarity and Structure:
The paper is clearly written and well-organized. The method is easy to follow, and the architecture is explained with sufficient diagrams and high-level intuition.

2. Improved Performance over Transfusion:
The method shows consistent improvements over Transfusion on language-only tasks as well as image understanding and image generation metrics.

Weaknesses:

1. Lack of conceptual novelty: The architectural design of separating modality-specific FFNs and QKV projections while sharing attention is highly derivative of existing multimodal systems(e.g. Flamingo[1]). Freezing base LLM is also a common practice in adaptation-based setups

2. Incomplete comparison set: The baseline comparison is limited to Transfusion. It omits stronger or more recent baselines like VILA-U[2], Janus-Pro[3], MUSE-VL[4] and so on.

3. Weak Evaluation Scope: The paper does not report any results of LMFusion on widely-used multimodal benchmarks such as MMMU, VQAv2, or GQA. These are essential for validating real-world image understanding performance, particularly for academic credibility. For image generation, it only reports on MS COCO and lack results on common text-to-image generation benchmark like GenEval. The paper also lacks a systematic discussion about related specialist models like common large multimodal models and diffsuion-based generative models.

4. Weak analysis for LLavaFusion:
The extension of LMFusion to LLaVA (LLavaFusion) is only briefly mentioned, with no substantial analysis and qualitative examples.

[1] Flamingo: a Visual Language Model for Few-Shot Learning. NeurIPS 2022.

[2] VILA-U: a Unified Foundation Model Integrating Visual Understanding and Generation. ICLR 2025.

[3] Janus-Pro: Unified Multimodal Understanding and Generation with Data and Model Scaling. Arxiv 2024.

[4] MUSE-VL: Modeling Unified VLM through Semantic Discrete Encoding. Arxiv 2024.

---

> ### Author Rebuttal · Authors · 2025-07-31
>
> We thank the reviewer for your time and constructive feedback! We address each of your comments below:
>
> > Lack of conceptual novelty: The architectural design of separating modality-specific FFNs and QKV projections while sharing attention is highly derivative of existing multimodal systems(e.g. Flamingo[1]). Freezing base LLM is also a common practice in adaptation-based setups
>
> While we agree that our work builds on established principles like freezing the base LLM, we believe our contribution is distinct from prior methods and offers a novel, systematic analysis.
>
> - **Architectural Distinction from Flamingo:** Models like Flamingo insert new cross-attention layers into a frozen LLM to incorporate final visual features output from a separate vision encoder. Our approach is different. We employ deep fusion, where each LLM layer can attend to every layer of the T2I model, enabling deep interactions between the T2I model and LLM.
>
> - **Systematic Analysis of Modality Separation:** Our core contribution is not just the final architecture, but the systematic investigation into the importance of modality separation when adapting a pretrained LLM for unified, diffusion-based generation. As detailed in our ablation studies (Section 5.1), we explicitly compare three distinct designs: no separation, shallow separation and deep separation.
>
> > Weak Evaluation Scope: The paper does not report any results of LMFusion on widely-used multimodal benchmarks such as MMMU, VQAv2, or GQA.
>
> We appreciate these suggestions for broader evaluation. In this paper, we focus on demonstrating LMFusion's core capabilities: image generation, preservation of language capabilities, and modality separation, with controlled comparison to Transfusion. While we provide preliminary image understanding results for LMFusion, we demonstrate the framework's broader applicability through LLaVAFusion (Table 2), which achieves highly competitive performance on comprehensive image understanding benchmarks, validating the effectiveness of our modality-specific design. Additionally, our method has been successfully inherited and applied to create real-world deployable models like Bagel [1] when trained on more comprehensive datasets, further demonstrating the practical value of our approach. We believe these results collectively validate both the technical soundness and real-world applicability of the LMFusion framework."
>
>
> > Weak analysis for LLavaFusion: The extension of LMFusion to LLaVA (LLavaFusion) is only briefly mentioned, with no substantial analysis and qualitative examples.
>
> Due to the page limit, we didn't include qualitative examples of LlavaFusion and focused on the controlled study of the LMFusion framework. Thanks for the suggestions. We will include them in the next version of the paper.
>
> > Could you share how LLaVAfusion performs on standard image understanding benchmarks (e.g., VQAv2, MMMU) compared to the base LLaVA-Next-8B? Additionally, while not directly claimed in the paper, I'm curious: Have you observed any effects—positive or negative—of jointly training for both image generation and image understanding on the model's performance in understanding tasks? Any insight or ablation in this direction would help clarify the broader implications of your approach.
>
> Our LLaVAFusion model is designed to fully preserve the understanding capabilities of the base LLaVA-Next-8B model. As shown in Table 2, the performance is identical to LLaVA-Next-8B due to our deep modality separation method, a key advantage of our framework.
>
> In our LMFusion experiments, we find that the vanilla design without modality separation would have image generation and image understanding compete for capacity. With the deep modality separation, with the same FLOPs, the compete problem disappears. We cannot include a link to new figures in the rebuttal, but compared to the existing Figure 4, the figure of deep modality separation under the same setup is significantly better. We will include it in the next version. For our LlavaFusion experiments, we have not yet explored joint training for image generation and understanding tasks. This is a very interesting future direction and we will include it in the revised version of the paper.
>
> > The paper observes that naive finetuning of dense pretrained LLMs for multimodal generation compromises their language capabilities, which motivates the modality separation approach in LMFusion. However, this might also suggest that the proposed architecture—by strictly separating modalities and freezing the language model—does not fully align image and text representations in a shared semantic space. How do the authors view this trade-off between preserving language ability and achieving deep cross-modal alignment? Are there any signs that the current setup limits joint reasoning or compositional understanding across modalities?
>
> Our modality-specific routing ensures that while the parameters remain separate, the representations can still interact meaningfully through the attention mechanisms. The frozen text modules provide a **stable semantic anchor**, while **the trainable image modules learn to align with this established space.** This design preserves the rich linguistic knowledge encoded in the pretrained LLM while enabling effective cross-modal understanding.
>
> > How the proposed approach compares conceptually and practically to specialist large multimodal models (e.g., Qwen-2.5-VL, GPT4o, Gemini) in terms of image-text understanding and joint reasoning.
>
> Our framework is designed to be backbone-agnostic and can incorporate any VLM as its foundation. While we experimented with LLaVA in this paper, the approach can be extended to other open-source VLMs such as Qwen-2.5-VL. This flexibility allows our method to leverage the strongest available multimodal understanding capabilities while adding unified generation functionality. Though one limitation would be that the base multimodal VLM needs to be open-source in our case, making extensions to GPT4o/Gemini difficult.
>
> > How LMFusion's diffusion-based image generation compares to specialized diffusion models (e.g., Flux, Stable diffusion 3), both in terms of quality and flexibility.
>
> LMFusion offers greater flexibility compared to specialized diffusion models like Flux and Stable Diffusion 3, as evidenced by our ability to lead to truly interleaved/omni models like Bagel. However, a direct comparison of image generation quality would be challenging due to differences in training data, a factor that significantly impacts model performance in real-world applications. Specialized diffusion models are typically trained on curated, high-quality image datasets optimized specifically for generation, while our approach prioritizes the integration of generation capabilities within a unified multimodal framework.
>
> We hope our responses have addressed the reviewer's concerns. Please let us know if you may have any questions.
>
> [1] Deng et al., 2025. Emerging Properties in Unified Multimodal Pretraining. https://arxiv.org/abs/2505.14683

---

> ### Author Response · Authors · 2025-08-04
>
> Thank you for the feedback and suggestion. Please let us know if you have any further questions. We would be happy to address them.

---

> ### Comment · Reviewer_FnMX · 2025-08-05
>
> I appreciate the authors' efforts for the rebuttal. I have some following questions:
>
> 1. Could authors share some comments about my Q2 for comparing with other recent unified methods?
>
> 2. I don't understand why LLaVAfusion performs same with llava-next-8B on image understanding tasks. According to Figure 1, the transformer of LM/VLM would accept an additional inputs from image. Even if the LM/VLM weights remain frozen, this additional inputs should still affect the LM/VLM's output. Do I understand it correctly?

---

> > ### Author Response · Authors · 2025-08-06
> >
> > Thank you for the follow-up questions. We address each question below:
> >
> > > Could authors share some comments about my Q2 for comparing with other recent unified methods? (VILA-U[2], Janus-Pro[3], MUSE-VL[4])
> >
> > We thank the reviewer for pointing to these relevant works. Despite being concurrent work to some of these more recent works, to provide a direct comparison, we have conducted new experiments with an 8-billion parameter version of our model, LMFusion-8B, using PerceptionLM as the frozen LM backbone [4]. The preliminary results are summarized below, comparing LMFusion-8B against recent unified models on key image generation and understanding benchmarks.
> > | Model            | DPGBench [1] | MMMU [2] | SEED [3] | POPE [4] |
> > |------------------|--------------|----------|----------|----------|
> > | **LMFusion 8B** (PLM backbone) | 77.12        | 46.1     | 79.3     | 89.9     |
> > | **Janus-Pro 7B**               | 76.45        | 41       | 72.1     | 87.4     |
> > | **VILA-U 7B**                  | -            | -        | 59       | 85.3     |
> > | **MUSE-VL 7B**                 | -            | 39.7     | 68.1     | -        |
> >
> > *The image generation for different models is evaluated under the same resolution, inference steps, CFG, etc. Due to time constraints during the rebuttal period, we were unable to run all models on the image generation benchmarks. We will include an expanded discussion on these models in the final version of the paper.
> >
> > As shown, our model performs better than Janus-Pro on image generation and understanding.
> >
> > More importantly, we hope to clarify that the primary focus of our paper is to provide a controlled comparison of model architectures/methodology under fixed data and compute budgets (e.g., comparing to the original Transfusion recipe). SOTA performance often depends heavily on the quality and quantity of the training data. For instance, Janus-Pro [3] uses their own so-called "normal text-to-image data" (no other details) and also 72 million samples of in-house synthetic data, which make a controlled comparison with our method challenging.
> >
> > > I don't understand why LLaVAfusion performs same with llava-next-8B on image understanding tasks.
> >
> > According to Figure 1, the transformer of LM/VLM would accept an additional inputs from image. Even if the LM/VLM weights remain frozen, this additional inputs should still affect the LM/VLM's output. Do I understand it correctly?
> >
> > When using llava-next as the backbone, the VLM inherently supports image inputs. By freezing the VLM parameters, we preserve the original model performance. Specifically,
> > When the LM module does not have a dedicated vision encoder: The UNet downsampler handles both image understanding and image generation tasks.
> > When the LM module has a dedicated vision encoder (llava): The UNet downsampler focuses solely on image generation, while image understanding is delegated to the frozen modules (shown in blue in Figure 1).
> > We thank the reviewer for raising this important question. We will provide clarification in the final version of the paper.
> >
> > [1] ELLA: Equip Diffusion Models with LLM for Enhanced Semantic Alignment
> >
> > [2] MMMU: A Massive Multi-discipline Multimodal Understanding and Reasoning Benchmark for Expert AGI
> >
> > [3] SEED-Bench: Benchmarking Multimodal LLMs with Generative Comprehension
> >
> > [4] PerceptionLM: Open-Access Data and Models for Detailed Visual Understanding

---

> > > ### Comment · Reviewer_FnMX · 2025-08-06
> > >
> > > Thanks for your quick reply and your clarification. Can you further clarify why PerceptionLM was chosen as the LM backbone, rather than LLaMA 3, as used in your paper?

---

> ### Author Response · Authors · 2025-08-06
>
> > Can you further clarify why PerceptionLM was chosen as the LM backbone, rather than LLaMA 3, as used in your paper?
>
> In our main paper experiments, we used the Llama architecture to maintain consistency with Transfusion, enabling controlled comparisons while preserving language understanding performance. For this evaluation, we selected PerceptionLM because our focus is to compare against SOTA image generation and understanding benchmarks. PerceptionLM's built-in image understanding capabilities make it more suitable for achieving the SOTA performance comparisons requested. This also ensures a fairer comparison, as other evaluated models prioritize image understanding over maintaining language-only performance.
>
> Additionally, PerceptionLM serves as a simple drop-in replacement within our framework. This compatibility further demonstrates the flexibility and adaptability of our proposed method.
>
> Thank you for the suggestion and question. Please let us know if you have any further questions. We would be happy to address them.

---

### Official Review · Reviewer_axpy · 2025-07-05

**Clarity:** 3
**Significance:** 3
**Originality:** 3
**Rating:** 5
**Confidence:** 4

**Summary:**

This paper proposes a framework called LMFusion, which aims to endow pre-trained text-only large language models (LLMs) with multimodal generation capabilities. The framework integrates pre-trained Llama models for language processing while introducing additional transformer modules to achieve visual understanding and generation capabilities. During the training process, the data from each modality is routed to its dedicated modules. Text-specific modules are frozen, and only image-specific modules are trained. Experimental results show that LMFusion outperforms Transfusion across various benchmarks while maintaining the language capabilities of Llama.

**Questions:**

- Please compare the training overhead of LMFusion and Transfusion to highlight the advantages of LMFusion.

**Ethical Concerns:**

["NO or VERY MINOR ethics concerns only"]

**Limitations:**

No major limitations and potential negative societal impact.

**Quality:**

3

**Strengths And Weaknesses:**

Strengths:

+ This paper is well-written and technically solid.
+ LMFusion retains the language processing capabilities of pre-trained LLMs by freezing text-specfic modules.
+ The experimental results show that, compared to naive fine-tuning pre-trained LLMs, LMFusion achieves significant improvements in image understanding and validates the effectiveness of the deep separation design scheme.

Weaknesses:

- LMFusion has twice as many parameters as Transfusion.
- LMFusion requires training image-specific modules from scratch, which incurs training overhead. It is unclear how much training overhead LMFusion can reduce compared to Transfusion.

---

> ### Author Rebuttal · Authors · 2025-07-31
>
> We thank the reviewer for your time and constructive feedback! We address each of your comments below:
>
> > LMFusion has twice as many parameters as Transfusion.
>
> While LMFusion has twice the total parameters of Transfusion, **it has the same training and inference FLOPs as Transfusion.** Our modality-specific routing ensures that for any given input token, only the corresponding modules are activated: text tokens are processed by the frozen text modules, and image tokens are processed by the trainable image modules.
>
> > LMFusion requires training image-specific modules from scratch, which incurs training overhead. It is unclear how much training overhead LMFusion can reduce compared to Transfusion.
>
> **LMFusion requires substantially less training overhead compared to Transfusion.** As shown in Figure 3, with less than one-third of Transfusion's training FLOPs, LMFusion achieves better performance than Transfusion across most image generation and understanding tasks while maintaining text performance.
>
> We hope our responses have addressed the reviewer's concerns. Please let us know if you may have any questions.

---

> ### Author Response · Authors · 2025-08-04
>
> Thank you for the feedback and suggestion. Please let us know if you have any further questions. We would be happy to address them.

---

### Decision · Program_Chairs · 2025-09-17

**Decision:**

Accept (poster)

**Comment:**

The paper presents LMFusion, a framework designed to equip pre-trained text-only large language models (LLMs), such as Llama-3, with multimodal generation capabilities, particularly for image generation and understanding.

Key Contributions:

•	Modular architecture: LMFusion introduces parallel transformer modules for image and text, initialized from the same LLM but trained independently.

•	Decoupled training: Text modules are frozen while image modules are trained, effectively preventing modality interference and language capability degradation.

•	Efficiency and performance: LMFusion is more compute-efficient than training multimodal models from scratch and achieves strong results across benchmarks.

•	Generalizability: The methodology is demonstrateded on both LLMs and existing VLMs.

Reviewers' overall attitudes:

•	most reviewers acknowledged the novelty, effectiveness, and forward-looking nature of the work.

•	The author’s rebuttal was well-received, resolving most concerns.

•	Most Reviewers emphasized the framework’s ability to retain language capabilities while adding image generation, marking it as a significant advancement in multimodal AI.